# Commercial integrated crop-livestock systems achieve comparable crop yields to specialized production systems: A meta-analysis

**Caitlin A. Peterson**[1], **Leonardo Deiss**[2], **Amélie C. M. Gaudin**[1]*

**1** Department of Plant Sciences, University of California, Davis, CA, United States of America, **2** College of Food, Agricultural, and Environmental Sciences, The Ohio State University, Columbus, OH, United States of America

* agaudin@ucdavis.edu

**Data Availability Statement:** The full dataset and accompanying R code are available on the Dryad repository with the following DOI: 10.25338/B8NP6J.

## Abstract

Production systems that feature temporal and spatial integration of crop and livestock enterprises, also known as integrated crop-livestock systems (ICLS), have the potential to intensify production on cultivated lands and foster resilience to the effects of climate change without proportional increases in environmental impacts. Yet, crop production outcomes following livestock grazing across environments and management scenarios remain uncertain and a potential barrier to adoption, as producers worry about the effects of livestock activity on the agronomic quality of their land. To determine likely production outcomes across ICLS and to identify the most important moderating variables governing those outcomes, we performed a meta-analysis of 66 studies comparing crop yields in ICLS to yields in unintegrated controls across 3 continents, 12 crops, and 4 livestock species. We found that annual cash crops in ICLS averaged similar yields (-7% to +2%) to crops in comparable unintegrated systems. The exception was dual-purpose crops (crops managed simultaneously for grazing and grain production), which yielded 20% less on average than single-purpose crops in the studies examined. When dual-purpose cropping systems were excluded from the analysis, crops in ICLS yielded more than in unintegrated systems in loamy soils and achieved equal yields in most other settings, suggesting that areas of intermediate soil texture may represent a "sweet-spot" for ICLS implementation. This meta-analysis represents the first quantitative synthesis of the crop production outcomes of ICLS and demonstrates the need for further investigation into the conditions and management scenarios under which ICLS can be successfully implemented.

## Introduction

Historically, agricultural systems produced a diverse set of plant and animal commodities by exploiting tight linkages between animals and crops to create closed or nearly closed resource loops. Outputs or waste material from one enterprise would often serve as a needed input for another enterprise, such as when crop residue was used to feed livestock and livestock manure

**Funding:** This study is based on work supported by the National Science Foundation Graduate Research Fellowship Program under grant number #1650042 (PI: P Mohapatra), an international travel allowance through the NSF-CAPES Graduate Research Opportunities Worldwide program to CP, and by the USDA National Institute of Food and Agriculture Experiment Station Hatch Projects to AG (project CA-D-PLS-2332-352H). NSF website: https://www.nsfgrfp.org/ USDA website: https://nifa.usda.gov/ The funders had no role in study design, data collection and analysis, decision to publish, or preparation of the manuscript.

**Competing interests:** The authors have declared that no competing interests exist.

was used to fertilize crops, creating a circular or semi-circular flow of nutrients and energy. Such mixed or integrated systems are still dominant in many traditional and smallholder settings globally [1]. However, they have been in sharp decline wherever agriculture has become industrialized due to a combination of cross-scale political, environmental, and socio-economic factors [2].

Specialized agricultural production systems are extremely efficient and productive, but they often come with environmental externalities. Specialization in crop enterprises can generate nutrient deficits and imbalances in nutrient cycling, leading to losses and/or inefficiencies that must be corrected with external inputs [3]. Specialized, intensive livestock enterprises such as industrial dairies and concentrated animal feeding operations create nutrient excesses leading to storage, disposal, and pollution problems [4]. Feed production for such operations creates further demand for the products of low-diversity corn and alfalfa systems and their associated consumption of valuable water resources [2]. These externalities are not limited to intensive systems; specialized extensive livestock enterprises such as grazed beef production create concerns over conversion of native habitat to pasture, e.g., in the Amazon, Cerrado, and Pampa ecosystems in Brazil and Argentina [5,6]. Unintended consequences of specialization and consolidation in agricultural landscapes also span the social and economic dimensions of food systems, including food insecurity–as when rural-to-urban displacement exacerbates poverty issues–and vulnerability of rural livelihoods to weather and market fluctuations due to dependence on fewer agricultural commodities [7,8].

Diversified cropping systems that re-integrate animal and crop enterprises, known as integrated crop-livestock systems (ICLS), are receiving renewed interest in large- and medium-scale operations [9–12]. They are an integral part of ecological intensification strategies, which aim to replace a portion of anthropogenic inputs with services from enhanced ecosystem functioning [13]. Producer motivations for re-integrating animals into cropland are varied, but often include risk reduction through diversification, increased nutrient and land-use efficiency, and climate resilience through enhanced adaptability of management options [14]. Commercially oriented ICLS can include systems such as sheep-vineyard operations in New Zealand [15], dual-purpose wheat in the southern Great Plains of the U.S. [16], and grazing of annual grass cover crops in Brazilian soybean systems [17]. They can also range in scale from the field to the territory level [18] and range in scope from relatively independent but cooperative enterprises [12] to systems that are fully integrated spatially, temporally, and managerially [19]. Each of these ICLS modalities features different levels of interaction among crop and livestock components, whether spatial, biological, temporal, or economic.

Integrated crop-livestock systems are also common and often the default system in the smallholder setting, where livestock feature as one in a number of small, diversified enterprises within a farm. Cut-and-carry systems are one example, where crop residues are harvested for livestock fodder and livestock manure is transported to fields for soil amendment. However, the objectives of these systems differ fundamentally from those of commercialized systems, in that subsistence is the goal rather than maximizing yield or revenue. Furthermore, the co-localization of the crop and livestock elements that characterize the systems examined here is often absent in the smallholder setting. Therefore, in this study we will focus exclusively on ICLS in the commercial, large- or medium-scale setting so as to be able to make comparisons among ICLS modalities.

Co-located ICLS, i.e., those that use the same land area for both crop and livestock production, are among the most ecologically complex ICLS modality in the commercial production setting. These include ICLS that synchronize crop and livestock production (e.g., dual-purpose crops, which are crops managed both for grazing and grain production) or that rotate crop and livestock components across seasons (e.g., forage or sod-based rotations and cover crop grazing) in a strategic manner. By introducing grazing animals into cropland or crops into

grazed systems, co-located ICLS create networks of interactions among plant, animal, and soil components that differ from simplified systems in sometimes unexpected ways [20]. These interactions create room for compensatory and synergistic processes that, in certain contexts, can improve productivity [16], environmental performance [21], and process- and system-level resilience to climate or market disturbances [22]. Process-level resilience, for example, can occur when cover crop grazing increases soil water use during the off season, but improves water availability for crop uptake in subsequent cropping seasons or under drought stress via grazing-driven improvements in soil physical and hydrological properties [23]. Amelioration of soil pH [24] and improvement in soil microbial activity and biomass production [25,26] in ICLS and grazing systems, especially in well weathered soils and no-till systems, are other examples of soil-driven processes impacting resilience. Promotion of system-level resilience includes cases where livestock integration decreases the volatility of farm incomes by capitalizing on opposing production trends of crop and livestock commodities [19].

While the socioeconomic benefits and tradeoffs affecting producers that implement crop-livestock integration have been well documented [27–29], reports on the effects of ICLS on process-level productivity and biophysical characteristics are often contradictory. The ICLS literature contains examples of increases [9] decreases [30], and no change [31] in subsequent crop yield with crop-livestock integration. Productivity outcomes are highly context specific and depend on interactions between management decisions and soil- and climate-related factors, and little is known as to how much environmental factors such as soil type, climate, and management strategy (ICLS modality) may predispose ICLS to success or failure in a given location. There is a need to understand the extent to which these moderating variables influence ICLS outcomes to determine the likelihood of their sustainability and performance in different regional environmental contexts.

This study represents the first time, to our knowledge, that crop production outcomes in ICLS have been examined across biogeographic regions and management scenarios. We performed a systematic review and meta-analysis of the literature based on eligibility criteria pertaining to four types of mechanized, annual row-crop ICLS: cover crop grazing, dual-purpose crops, forage rotations, and stubble grazing. Our objectives were to: 1) understand the effect of crop-livestock integration on crop productivity under normal and abnormally dry weather conditions, 2) determine likely production outcomes across environments and ICLS types, and 3) identify the most important variables related to production outcomes for each type of ICLS. For each ICLS type, we compared the crop yield response of integrated (grazed) treatments to an unintegrated (ungrazed) control, along with potential moderating variables such as crop type, animal type, climate, the occurrence of weather anomalies, and soil characteristics. Our results inform the role of ICLS in agricultural adaptation to climate change, sustainable production systems, and ecological intensification of agriculture.

## Methods

### Identification of studies

We conducted a comprehensive literature search using three academic databases (Web of Science, CAB Abstracts, and Agricola) and the Google Scholar internet search engine in English, French, Spanish, and Portuguese. The most recent database search was conducted in September 2018. We gleaned further records from the reference lists of review articles and research articles meeting the initial eligibility criteria. Targeted searches of governmental and independent agricultural research organizations were also performed in countries where medium-to-large scale, commercially oriented ICLS are known to occur. Finally, we performed a manual search of the grey literature including theses and dissertations and data from long-term

experiments, both published and unpublished, in consultation with prominent integrated crop-livestock system researchers.

No prior review protocol existed for this study. The following search terms were employed for abstracts, titles, and keywords: (crop-livestock AND yield) NOT mixed); ("crop-livestock" AND integ*) OR "integração lavoura-pecuária" OR "integración agropecuaria"; "crop-live-stock" AND yield; (crop*livestock OR crop OR livestock) AND (French OR France) AND yield AND graz*; (crop*livestock OR crop OR livestock) AND (Spain OR Spanish OR "Latin America" OR "South America") AND yield AND graz*; intégration ("polyculture-élevage" OR polyculture OR élevage OR agriculture) rendement pâturage expérimental -arbres. Search results were deduplicated and restricted to full-text journal articles. Google Scholar results were additionally restricted to the years 2008–2018 due to the volume of results; other databases were searched for the full range of available years.

A total of 2,702 studies were identified from the database searches, unpublished dissertations, reference lists of eligible studies and literature reviews, and long-term datasets provided by ICLS researchers (Fig 1). The initial screening process involved manual scanning of titles and abstracts for clear instances of ineligibility, e.g., wrong field of study, wrong scope, wrong subject, or wrong language. A total of 2,569 records were excluded in the initial screening process, leaving the full text of 133 articles to be assessed in greater detail based on the following eligibility criteria:

1. Study scope was restricted to agropastoral systems with annual crops. Duck-rice-azolla, agro-silvo-pastoral systems, and systems integrating livestock with perennial crops were excluded;

2. Study involved a replicated field trial with both an integrated system (grazed treatment) and an unintegrated control (ungrazed treatment) and included at least one season each of the cropping component and the grazing component;

3. Crops and livestock were co-located, i.e., spatially integrated at the field level. Cut-and-carry, manure amendments, or farm-level mixed systems were excluded due to disparities in system objectives and constraints as well as difficulties in determining adequate experimental controls for farm-level integration;

4. Study was original research, dataset, or dissertation, i.e., not a review, book chapter, or conference proceeding.

Sixty-six studies met our criteria for inclusion in the meta-analysis, two of which included unpublished data. These studies yielded a total of 246 individual observations, spanning 3 continents, 6 countries, 12 crops, and 4 livestock types (beef cattle, dairy cattle, sheep, and goats) (Table 1). Four types of ICLS were identified: 1) forage rotation, or a multi-year rotation of crops with semi-permanent pasture or turf grazed by livestock (also known as sod-based rotations); 2) cover crop grazing, or an annual rotation of a cash crop with an off-season grazed forage; 3) stubble grazing, or livestock grazing of crop residues left over after harvest; 4) dual-purpose crops, or crops that are grazed by livestock in early phenological stages and subsequently allowed to mature for grain harvest (Table 2). These ICLS categories are distinct from farm-level or territory-level integration of crops and livestock, e.g., through cutting forage from one part of the farm and transporting it to livestock feeding locations on other parts of the farm (farm-level integration), or through cooperation among farming operations within a territory to supply needed inputs, such as when livestock operations supply manure to separate crop operations for fertility treatments (territory-level integration).

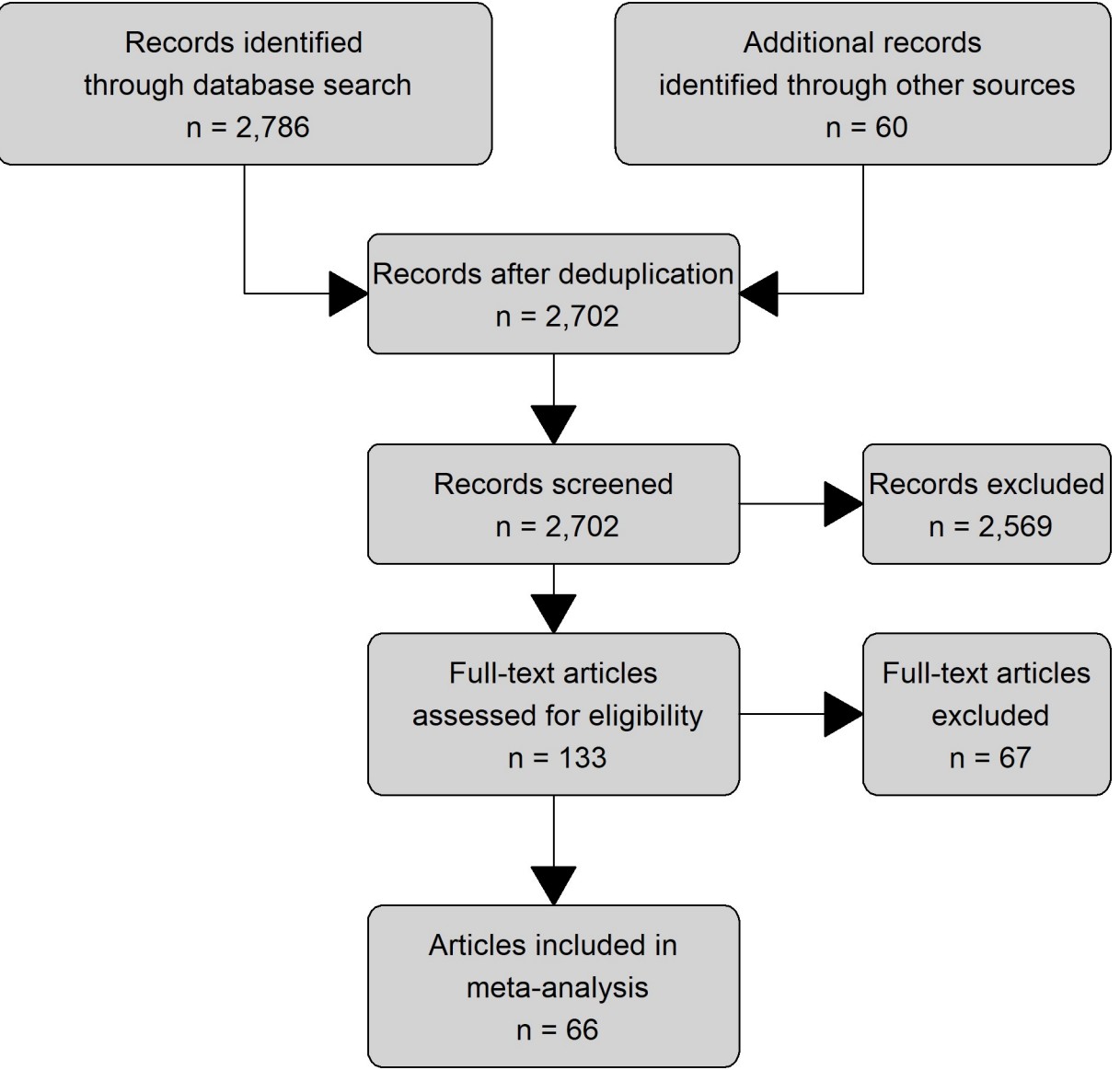

**Fig 1. PRISMA diagram of the study selection process.** Diagram shows the number of studies excluded at each screening step. Adapted from Moher et al. [32].

Geographically, studies were concentrated in North America, South America, and Australia (Fig 2A), likely due to the greater prevalence of co-located ICLS involving annual crops in both the research and production sectors in these regions. In Africa, Europe, and Southeast Asia, ICLS often take the form of farm-level or territory-level integration or involve perennial rather than annual crops [95]. Disparities in terminology and experimental design may also have caused ICLS studies from these regions to be excluded.

Data for yields of the crop following a grazing treatment, whether in rotation or later in the same season, were collected for each study, in addition to information on experimental design, length of the experiment, and estimated within-treatment error where available. The necessary data were sought in the methods, results, and tables and figures. When data were given only in figures, values were extracted using the image calibration plugin in the ImageJ image

**Table 1. Summary of the studies included in the meta-analysis and key environmental moderating variables.**

| Authors | State, country | Study length (yrs.) | No. observations | ICLS type | Crop(s) | Animal | Avg. precip. (mm yr$^{-1}$) | Avg. temp. (°C) | Soil texture |
|---|---|---|---|---|---|---|---|---|---|
| Agostini et al. [33] | BA, AR | 1 | 1 | stubble grazing | corn | beef | 875 | 15.7 | light clay |
| Allen et al. [34] | TX, US | 5 | 8 | forage rotation | cotton | beef | 484 | 14.8 | loam |
| Allen et al. [35] | TX, US | 8 | 5 | forage rotation | cotton | beef | 484 | 14.8 | loam |
| Assmann et al. [36] | PR, BR | 1 | 1 | cover crop grazing | corn | beef | 1,737 | 15.7 | sandy clay loam |
| Assmann et al. [37] | PR, BR | 1 | 1 | cover crop grazing | soybean | beef | 1,907 | 17.4 | loamy sand |
| Balbinot Junior et al. [38] | SC, BR | 4 | 4 | cover crop grazing | bean | dairy | 1,495 | 16.3 | clay loam |
| Balbinot Junior et al. [39] | SC, BR | 5 | 5 | cover crop grazing | bean | dairy | 1,495 | 16.3 | clay loam |
| Bartmeyer et al. [40] | PR, BR | 1 | 1 | dual-purpose crop | wheat | beef | 1,539 | 16.4 | sandy loam |
| Baumhardt et al. [41] | TX, US | 5 | 9 | stubble grazing | wheat | beef | 477 | 13.6 | loam |
| Baumhardt et al. [42] | TX, US | 10 | 14 | stubble grazing | sorghum | beef | 477 | 13.6 | loam |
| Bell et al. [43] | NSW, AU | 1 | 2 | stubble grazing | wheat, canola | sheep | 747 | 16.0 | loamy sand |
| Bonetti et al. [44] | GO, BR | 1 | 1 | cover crop grazing | soybean | beef | 1,539 | 22.7 | loamy sand |
| Bortolini et al. [45] | PR, BR | 1 | 2 | dual-purpose crop | oat | beef | 1,737 | 15.7 | sandy clay loam |
| Carvalho et al.* | RS, BR | 14 | 14 | cover crop grazing | soybean | beef | 1,824 | 19.5 | loamy sand |
| Carvalho et al.* | RS, BR | 3 | 6 | cover crop grazing | corn, soybean | sheep | 1,358 | 20.0 | light clay |
| Christiansen et al. [46] | OK, US | 3 | 3 | dual-purpose crop | wheat | beef | 837 | 15.3 | loam |
| Cicek et al. [47] | MB, CA | 3 | 4 | cover crop grazing | rye, wheat | sheep | 491 | 2.3 | loam |
| Clark et al. [48] | IA, US | 3 | 3 | stubble grazing | soybean | beef | 864 | 9.0 | loamy sand |
| Dann et al. [49] | ACT, AU | 2 | 18 | dual-purpose crop | rye, wheat, oat, barley | sheep | 681 | 13.9 | silt loam |
| Debiasi and Franchini [50] | PR, BR | 1 | 2 | cover crop grazing | soybean | beef | 1,437 | 20.1 | loamy sand |
| Edwards et al. [51] | OK, US | 3 | 3 | dual-purpose crop | wheat | beef | 837 | 15.3 | loam |
| Ferreira et al. [52] | PR, BR | 2 | 15 | cover crop grazing | soybean | beef | 1,411 | 20.2 | sandy loam |
| Franchin et al. [53] | PR, BR | 1 | 2 | cover crop grazing | corn | dairy | 1,797 | 17.3 | loamy sand |
| Franchini et al. [54] | PR, BR | 1 | 1 | cover crop grazing | soybean | beef | 1,437 | 20.1 | loamy sand |
| Franzluebbers and Stuedemann [55] | GA, US | 4 | 1 | cover crop grazing | sorghum, corn, wheat | beef | 1,298 | 15.8 | loamy sand |
| Franzluebbers and Stuedemann [56] | GA, US | 8 | 8 | cover crop grazing | corn, soybean, wheat | beef | 1,298 | 15.8 | loamy sand |

*(Continued)*

**Table 1.** (*Continued*)

| Authors | State, country | Study length (yrs.) | No. observations | ICLS type | Crop(s) | Animal | Avg. precip. (mm yr⁻¹) | Avg. temp. (°C) | Soil texture |
|---|---|---|---|---|---|---|---|---|---|
| George et al. [57] | FL, US | 4 | 7 | cover crop grazing | peanut, cotton | beef | 1,409 | 18.3 | loam |
| Harrison et al. [58] | ACT, AU | 2 | 2 | dual-purpose crop | wheat | sheep | 681 | 13.9 | silt loam |
| Hunt et al. [59] | NSW, AU | 4 | 7 | stubble grazing | wheat, barley, canola | sheep | 712 | 16.3 | silt loam |
| Kilcher [60] | SK, CA | 3 | 3 | dual-purpose crop | rye | beef | 343 | 3.3 | loamy sand |
| Kirkegaard et al. [61] | ACT, AU | 3 | 3 | dual-purpose crop | canola | sheep | 681 | 13.9 | silt loam |
| Kirkegaard et al. [62] | NSW, AU | 3 | 6 | dual-purpose crop | canola | sheep | 730 | 15.7 | loamy sand |
| Kirkegaard et al. [63] | NSW, AU | 1 | 3 | dual-purpose crop | canola | sheep | 730 | 15.7 | loamy sand |
| Kunz et al. [64] | RS, BR | 1 | 1 | cover crop grazing | soybean | beef | 1,656 | 19.8 | loamy sand |
| Lang et al. [65] | PR, BR | 1 | 1 | cover crop grazing | corn | beef | 1,499 | 17.4 | sandy loam |
| Lenssen et al. [66] | MT, US | 4 | 8 | stubble grazing | wheat | sheep | 430 | 7.8 | loamy sand |
| Loison et al. [67] | FL, US | 1 | 1 | forage rotation | cotton | beef | 1,409 | 18.3 | loam |
| Maughan et al. [68] | IL, US | 4 | 4 | forage rotation | corn | beef | 985 | 11.7 | loam |
| Miller et al. [30] | MT, USA | 7 | 11 | stubble grazing | wheat | sheep | 430 | 7.8 | loamy sand |
| Modolo et al. [69] | PR, BR | 1 | 1 | cover crop grazing | corn | beef | 1,797 | 17.3 | loamy sand |
| Moraes et al.* | PR, BR | 3 | 4 | cover crop grazing | corn | sheep | 1,737 | 15.7 | sandy clay loam |
| Nicoloso et al. [70] | RS, BR | 2 | 2 | cover crop grazing | soybean | beef | 1,794 | 19.2 | light clay |
| Novakowiski et al. [71] | PR, BR | 1 | 5 | cover crop grazing | corn | sheep | 1,737 | 15.7 | sandy clay loam |
| Pitta et al. [72] | PR, BR | 1 | 1 | dual-purpose crop | wheat | beef | 1,907 | 17.4 | loamy sand |
| Pitta et al. [73] | PR, BR | 1 | 1 | cover crop grazing | corn | goat | 1,911 | 17.4 | loamy sand |
| Proffitt et al. [74] | WA, AU | 1 | 1 | forage rotation | wheat | sheep | 323 | 19.2 | loam |
| Radford et al. [75] | QLD, AU | 3 | 2 | stubble grazing | wheat | beef | 699 | 18.6 | silt loam |
| Rakkar et al. [76] | NE, US | 11 | 11 | stubble grazing | corn, soybean | beef | 774 | 9.6 | loamy sand |
| Sandini [77] | PR, BR | 1 | 3 | cover crop grazing | bean | sheep | 1,737 | 15.7 | sandy clay loam |
| Santos et al. [78] | RS, BR | 6 | 6 | cover crop grazing | wheat | beef | 1,737 | 18.6 | light clay |
| Santos et al. [79] | RS, BR | 9 | 9 | cover crop grazing | wheat | beef | 1,737 | 18.6 | light clay |
| Schomberg et al. [80] | GA, US | 4 | 4 | cover crop grazing | cotton | beef | 1,256 | 16.1 | loamy sand |

(*Continued*)

**Table 1.** (Continued)

| Authors | State, country | Study length (yrs.) | No. observations | ICLS type | Crop(s) | Animal | Avg. precip. (mm yr⁻¹) | Avg. temp. (°C) | Soil texture |
|---|---|---|---|---|---|---|---|---|---|
| Shimoda et al. [81] | AP, PY | 3 | 4 | forage rotation | soybean, wheat | beef | 1,521 | 21.3 | light clay |
| Silva et al. [82] | PR, BR | 2 | 2 | cover crop grazing | soybean, corn | dairy | 1,737 | 15.7 | sandy clay loam |
| Silveira et al. [83] | GO, BR | 1 | 1 | forage rotation | bean | beef | 1,393 | 22.1 | sandy loam |
| Silveira et al. [84] | SC, BR | 1 | 1 | cover crop grazing | corn | dairy | 1,495 | 16.3 | clay loam |
| Sprague et al. [85] | ACT, AU | 2 | 4 | dual-purpose crop | canola | sheep | 681 | 13.9 | silt loam |
| Sprague et al. [86] | ACT, AU | 2 | 5 | dual-purpose crop | wheat, canola | sheep | 681 | 13.9 | silt loam |
| Stalker et al. [87] | NE, US | 5 | 5 | stubble grazing | corn | beef | 502 | 9.6 | silt loam |
| Taffarel et al. [88] | PR, BR | 2 | 6 | cover crop grazing | soybean | beef | 1,641 | 20.0 | loamy sand |
| Tanaka et al. [89] | ND, US | 4 | 10 | stubble grazing | corn, triticale, oat | beef | 418 | 5.3 | loam |
| Tracy and Zhang [90] | IL, US | 3 | 3 | forage rotation | corn | beef | 985 | 11.7 | loam |
| Trogello et al. [91] | PR, BR | 1 | 1 | cover crop grazing | corn | beef | 1,734 | 18.4 | loamy sand |
| Veiga et al. [92] | SC, BR | 3 | 2 | cover crop grazing | soybean, corn | beef | 1,766 | 16.6 | loamy sand |
| Veiga et al. [93] | SC, BR | 6 | 6 | cover crop grazing | soybean, corn | beef | 1,766 | 16.6 | loamy sand |
| Virgona et al. [94] | NSW, AU | 2 | 1 | dual-purpose crop | wheat | sheep | 730 | 15.7 | loamy sand |

*Data for multi-year experiments requested directly from lead investigator. Include both published and unpublished data.

processing software (v.1.5.2b). In cases where data were incomplete or estimates of error were not reported, investigators were contacted. When investigators could not be reached or the requested information was not available, standard deviations (SD) of crop yields were imputed

**Table 2. Summary of ICLS types and their control systems examined in the meta-analysis.**

| ICLS type | Description | Control system | Examples |
|---|---|---|---|
| Forage rotation | Multi-year rotation of annual crops with grazed forage crops, e.g., 3 years of grazed perennial ryegrass rotated with 2 years of peanuts and 1 year of corn. | Multi-year rotation of annual crops with ungrazed grass cover crop. Also known as ley cropping or sod-based rotation. | Loison et al. 2012 [57]; Allen et al. 2007 [35] |
| Cover crop grazing | Yearly rotation of a main season cash crop with a grazed, off-season cover crop or forage crop, e.g., summer soybean rotated yearly with winter grazed annual ryegrass pasture. | Yearly rotation of a main season cash crop with an ungrazed grass cover crop. | Moraes et al. 2014 [10]; Franzluebbers and Stuedemann 2014 [56] |
| Stubble grazing | Animal grazing of standing crop residue left over after harvest, e.g., supplemental grazing of corn stalks. | Stubble left in field with no grazing. | Baumhardt et al. 2011 [42]; Radford et al. 2008 [75] |
| Dual-purpose crop | Annual crop grazed in the vegetative stage and subsequently allowed to mature for grain harvest following the removal of grazing animals, e.g., early-stage wheat grazed by sheep and later harvested for grain. | Crop not grazed in early stages, harvested normally (single-purpose crop). | Kirkegaard et al. 2012 [63]; Edwards et al. 2010 [51] |

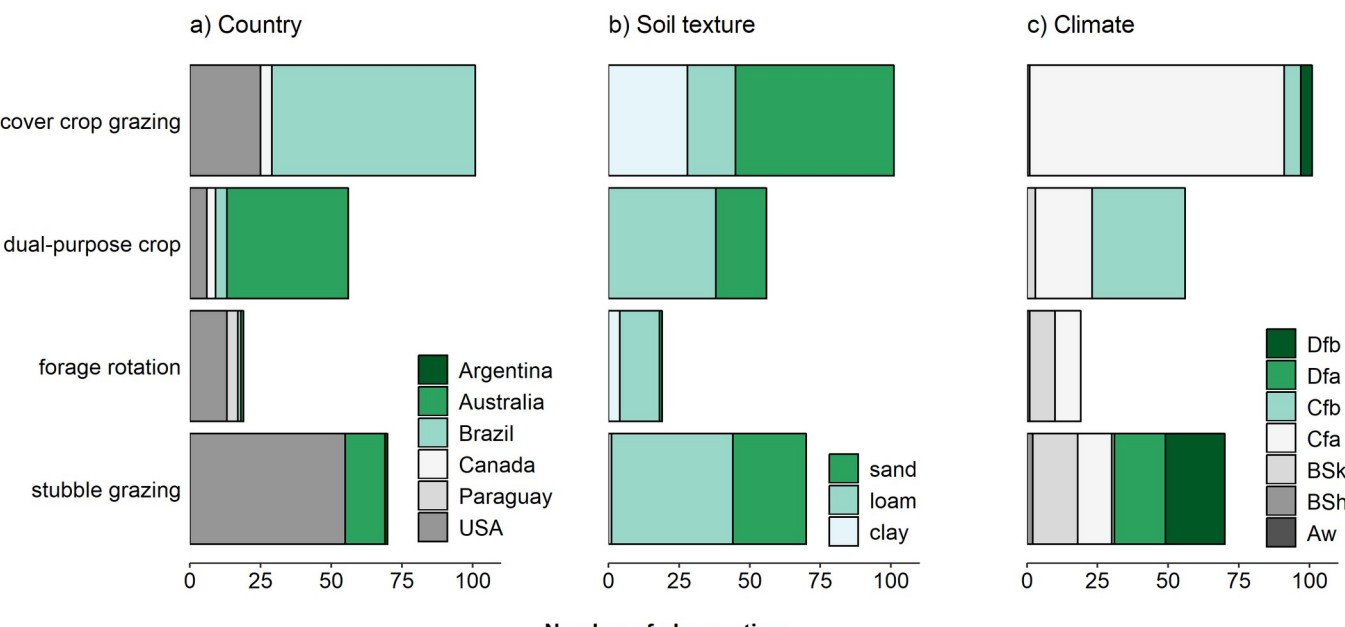

**Fig 2. Graphical summary of studies used in the meta-analysis.** Shown are the number of observations within Integrated Crop-Livestock System types by (A) country, (B) soil texture, and (C) climate. Köppen climate class abbreviations are as follows: Aw = tropical savannah; BSh = hot semi-arid; BSk = cold semi-arid; Cfa = humid subtropical; Cfb = temperate oceanic; Dfa = hot-summer humid continental; Dfb = warm-summer humid continental.

as in Eq 1:

$$\widetilde{SD}_j = \bar{X}_j \left( \frac{\sum_i^K SD_i}{\sum_i^K \bar{X}_i} \right) \tag{1}$$

where ~ indicates an imputed estimate, $\bar{X}_j$ is the observed mean of the study with the missing SD, and K is the number of jth studies with complete information [96]. This step was taken for 59 (or 24%) of the 246 total observations included in the analysis.

## Moderating variables

Data on categorical environmental moderating variables were also collected for each study (Table 1; Fig 2). Studies were grouped into climate and soil classes according to the Köppen climate classifications, which were extracted from the updated world map of the Köppen-Geiger climate [97], and soil texture characteristics extracted from the Harmonized World Soil Database v1.2 [98]. Additional moderating variables included crop species, livestock species, and the occurrence of dry weather anomalies. The latter was defined as a season during which precipitation accumulation was abnormally low according to specifications set by the authors of the relevant study. Crop species were grouped according to broad agronomic similarities: cereals (corn and sorghum), small grains (wheat, oat, barley, triticale, and rye), fiber (cotton harvested for lint), soybean, other legumes (peanuts and common bean), and oilseeds (canola). For animals, goats and sheep were grouped into small ruminants and beef and dairy cattle were grouped under cattle.

Productive outcomes in a given system are dependent on proper management of inputs and grazing animals. Regarding inputs, all studies involved the use of synthetic fertilizers and did not include application of manure as slurry or compost. However, optimal grazing rates or fertilizer application rates do not necessarily translate among different system contexts. In

particular, studies are inconsistent in their reporting of animal stocking densities and often report only mean densities when stocking is allowed to vary across the grazing season in accordance with forage availability. Similarly, fertilizer application rates varied among studies because experimental plots were managed according to soil test recommendations or based on yield goals. Given the above considerations, we assumed that the crops and pastures compared here were nutrient-unlimited and aligned with best nutrient and grazing management practices for a given environmental or managerial context. Where a study involved multi-level or crossed treatment designs for fertilizer application rates, tillage types, or grazing intensities/ stocking densities, only the treatment values closest to the best management practice for that system were used in the analysis. If more than one treatment value was commensurate with best management practices (based on the conclusions of study authors), we used the average result for all best treatment values.

## Statistical analysis

Analyses were carried out using the randomForest [99] and boot [100,101] packages in R v.3.5.2. Funnel plots were assessed visually for asymmetries that would indicate publication bias, and the "trim and fill" method [102] was used to determine if results were robust to adjustment for funnel plot asymmetry (S1 Fig). An Egger test for funnel plot asymmetry indicated no significant evidence of publication bias (z = 0.72, p = 0.47). In addition, Rosenberg's fail-safe number [103] indicated that more than 1,000 additional studies would be needed to significantly alter the effect size, a large enough number that any potential publication bias can safely be ignored in this analysis.

For each treatment-control pair, the effect size was calculated as the log response ratio (LRR) [**104**] given in Eq 2:

$$LRR = \ln\left(\frac{\bar{X}_{grazed}}{\bar{X}_{ungrazed}}\right) \tag{2}$$

Because it is a ratio, the LRR allows crop groups with different expected yield magnitudes to be compared directly. A positive LRR indicates that grazing animal integration increased crop yield relative to the ungrazed control system, while a negative LRR indicates that grazing animal integration decreased crop yield relative to the control. Weights were assigned to LRRs in proportion to the inverse of within-study variance. For studies that spanned multiple years or growing seasons, each season was treated as a separate observation. Observations were omitted when crop yields equaled zero or when yields were more than 5 standard deviations away from the weighted mean. This step resulted in the elimination of 4 observations from the dataset.

The overall mean effect size and 95% confidence intervals were determined using non-parametric, bias-adjusted-accelerated (BCa) bootstrapping procedures in the boot package of R with 4,999 iterations [105]. The overall effect was considered significantly positive or negative if the bootstrapped confidence intervals did not include zero. For interpretability, all results were back-transformed from the LRR to percent difference in yield between the grazed and control treatments.

A non-parametric random forest procedure was performed on a random effects model using the randomForest package in R to determine variable importance rankings [105]. The random effects model included LRR as the response variable and study, ICLS type, weather anomalies, crop type, animal type, climate, length of study, and soil texture as categorical predictor variables. Random permutation of input variables determined variable importance as the percent increase in mean squared error (MSE) when a given variable was removed from

the model (Type 1). An increase in MSE represented a decrease in model accuracy. Variable importance was normalized by setting the most important variable, or the variable with the highest percent increase in MSE, equal to 1 and calculating the importance of the remaining variables relative to the most important variable. Negative variable importance values represented variables that were no more informative to the model than random chance.

Mean effect sizes for subgroups within moderating variables were determined using the same bootstrapping procedure to better understand the effect of environmental and management variables on ICLS outcomes. Differences among subgroup categories within a moderating variable were assessed using randomization tests with 999 permutations. A one-sided p-value of <0.05 indicated a significant difference among subgroup categories. Subgroups with small sample sizes (n < 15 observations), which included tropical wet savannah and hot semi-arid climate subgroups, were excluded from subgroup analysis. Similarly, two studies (n = 11 observations) that involved irrigated system designs were excluded from the dry weather anomaly subgroup analysis. Interactions among subgroups were not tested due to sample size limitations for many interaction effects.

## Results

### Livestock integration in cropping systems: Overall effect

Integrated systems had similar yields to unintegrated systems, with a non-significant negative effect of -1% (and confidence interval of -7% to +2%) not considering ICLS category, crop category, or other moderating variables (Fig 3). While many of the studies included in the analysis observed slight yield effects from grazing treatments, most yield impacts were minor. Different categories of ICLS demonstrated no difference in yields between integrated treatments and unintegrated controls with the exception of dual-purpose cropping systems, where grazing led to significantly lower yields (-20%) on average than unintegrated, single-purpose controls. When observations from dual-purpose cropping systems were excluded, the overall effect size became slightly positive (+1%) but remained non-significant.

As expected given the large deviation in effect size for the dual-purpose category, the randomization test indicated more heterogeneity among categories of ICLS than would be expected by chance (p = 0.01). For subsequent subgroup analyses, we performed one analysis excluding observations from dual-purpose crop ICLS and a second analysis including these observations to account for the possible confounding effect of disproportionate representation of dual-purpose studies in certain crop and climate categories (e.g., canola and wheat crops, humid subtropical climates). Hereafter we present results only from the subgroup analyses where observations from dual-purpose cropping systems were excluded.

The random forest variable importance ranking procedure identified three moderating variables that had the most influence on model fit, i.e., were the most important predictors of effect size: 1) crop, 2) soil texture, and 3) the occurrence of in-season dry weather anomalies, with crop being the most important (Fig 4). Out-of-bag error, which in this case refers to the mean squared error of the test set, was 0.05, with percent increases in MSE ranging from -4 to 9.

### Priority moderating variables: Crops, soils, and weather anomalies

ICLS had no effect on yields for any crop category when dual-purpose crop observations were excluded from the bootstrap analysis (Fig 5A), and there was no significant heterogeneity among different crop categories (p = 0.2). When dual-purpose crop observations were included, ICLS had a significantly negative (-9%) effect on canola yields relative to unintegrated controls, as did ICLS in the small grains category (wheat, oat, barley, rye, and triticale;

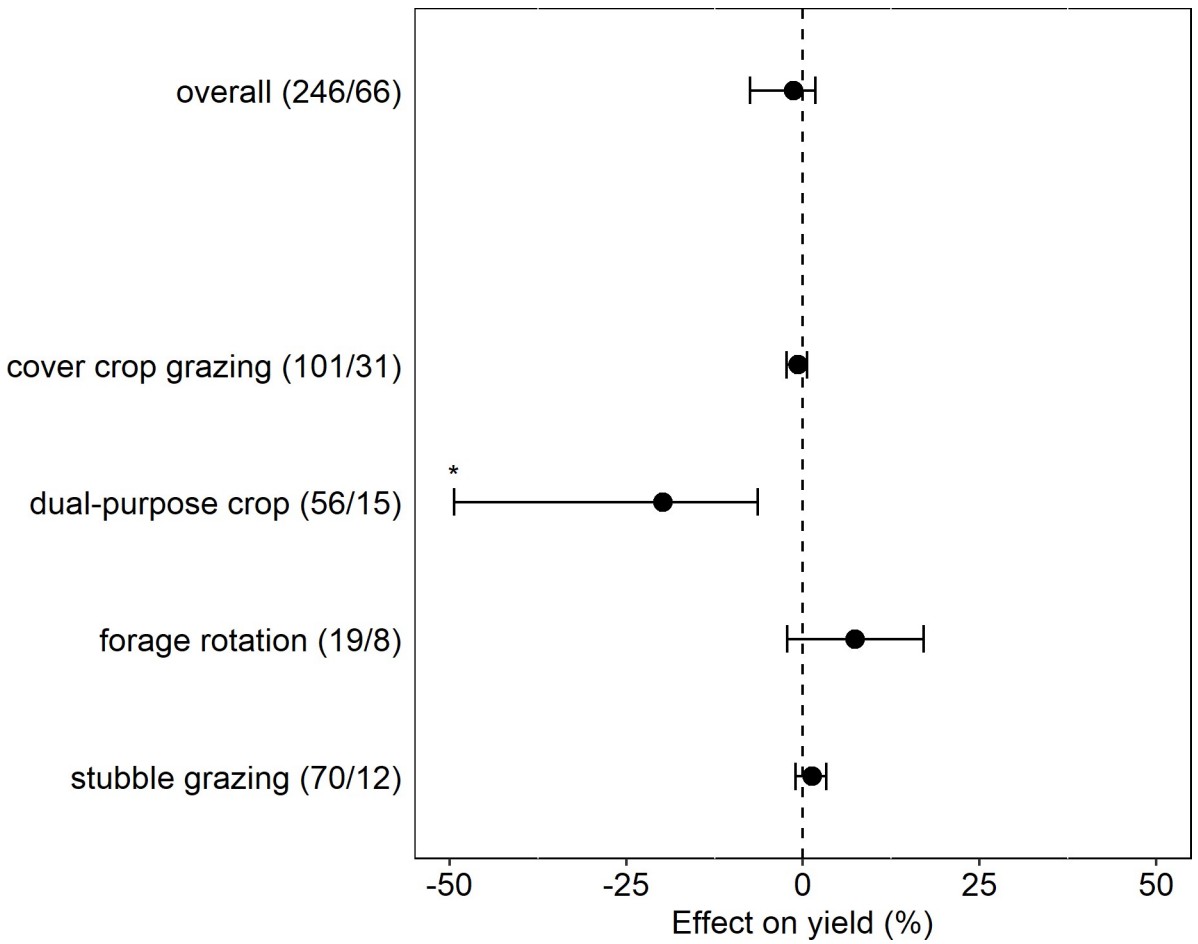

**Fig 3. Overall effect of ICLS on crop yield and effect within different ICLS system types.** Number of observations/number of studies for each category appears in parentheses. Points represent grazed system yield effect, while the dotted vertical line represents the paired ungrazed system yield. Error bars represent 95% bias-corrected-accelerated bootstrap confidence intervals. Asterisks (*) represent a significant yield response in grazed systems relative to ungrazed systems at the 95% confidence level.

-8%; S2A Fig), due to the disproportionate representation of these crops in dual-purpose systems.

ICLS implemented in loamy soil types had 5% higher yields than unintegrated systems, whereas there was no difference between integrated and unintegrated systems in clay and sand soils (Fig 5B). There were no significant differences among soil texture categories according to the randomization test (p = 0.5). When observations from dual-purpose crop systems were included, ICLS had a negative 4% impact on yields in sandy soils relative to unintegrated systems (S2B Fig).

Approximately 27% of studies reported abnormally dry weather during at least one recorded growing season, or 18 out of 66 total studies. Of those anomalously dry seasons, 47% occurred in cold semi-arid climates, 29% occurred in temperate oceanic climates, 4% occurred in warm-summer humid continental climates, and 18% occurred in humid subtropical climates. However, there was no significant effect of dry weather anomalies on yield in ICLS treatments relative to unintegrated controls. There was a non-significant negative effect during both anomalously dry years (-9%, n = 25) and normal precipitation years (-1%, n = 155), and no significant heterogeneity between the two categories (p = 0.3) (Fig 5C). This non-effect of

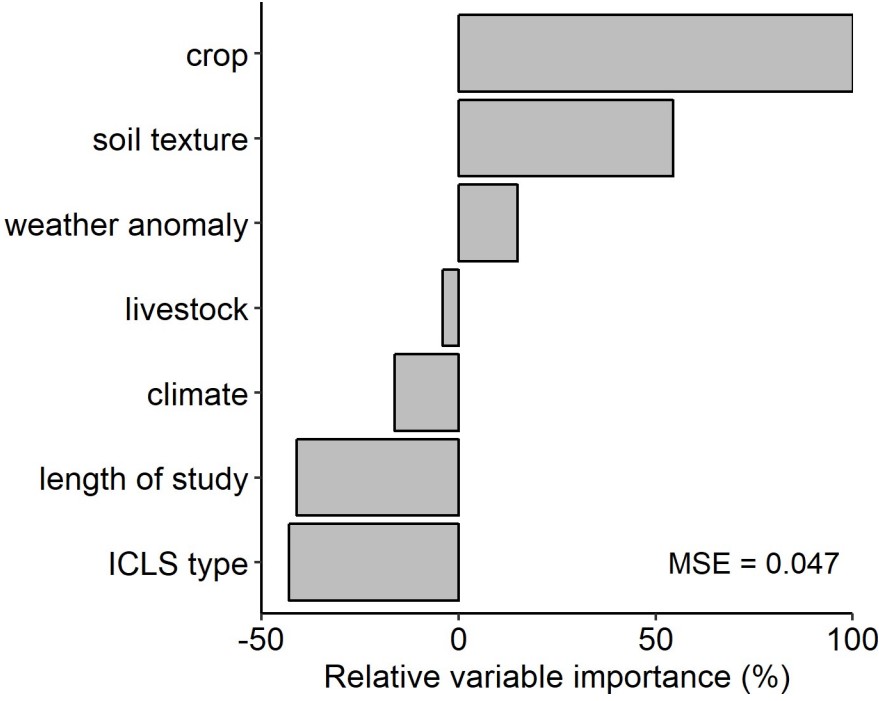

**Fig 4. Relative variable importance ranking.** Rankings were derived from a random forest permutation procedure, with the effect size of ICLS relative to unintegrated systems as the response variable. Model variables were ranked according to the percent increase in mean squared error (MSE, or out-of-bag error) when they were removed from the model (Type 1 classification) and normalized relative to the most important variable. Negative variable importance indicates variables that were no more informative to the model than random chance.

dry weather anomalies was upheld when observations from dual-purpose crop systems were included (S2C Fig).

## Minor moderating variables: Climate, livestock, and study conditions

ICLS had a positive effect (+9%) on yields in temperate oceanic climates, but confidence intervals for this category could not be generated due to the small sample size (n = 7; Fig 6A). There were no differences between integrated and unintegrated systems in other climates, and no significant heterogeneity among regional climate subgroup categories (p = 0.3). When dual-purpose crop observations were included, ICLS had a negative effect on yields in humid subtropical climates (-3%; n = 131; S3A Fig).

For all livestock categories (small ruminant and cattle), ICLS had similar yields to unintegrated controls (S4A and S4B Fig). There was no significant heterogeneity among livestock categories (p = 0.6). These results were unchanged when dual-purpose crop system observations were included.

Study duration had no bearing on the impact of ICLS on crop yields, and the randomization test indicated no significant differences among categories of study duration (1–2 yrs, 3–5 yrs, and 6 or more years; p = 0.3; Fig 6B). When observations from dual-purpose crop systems were included, studies of length 2 years or less showed a significantly negative (-5%) yield effect in ICLS relative to unintegrated controls, while longer studies showed no difference between integrated and unintegrated systems (S3B Fig). Again, this result can be attributed to the shorter duration on average of studies involving dual-purpose systems.

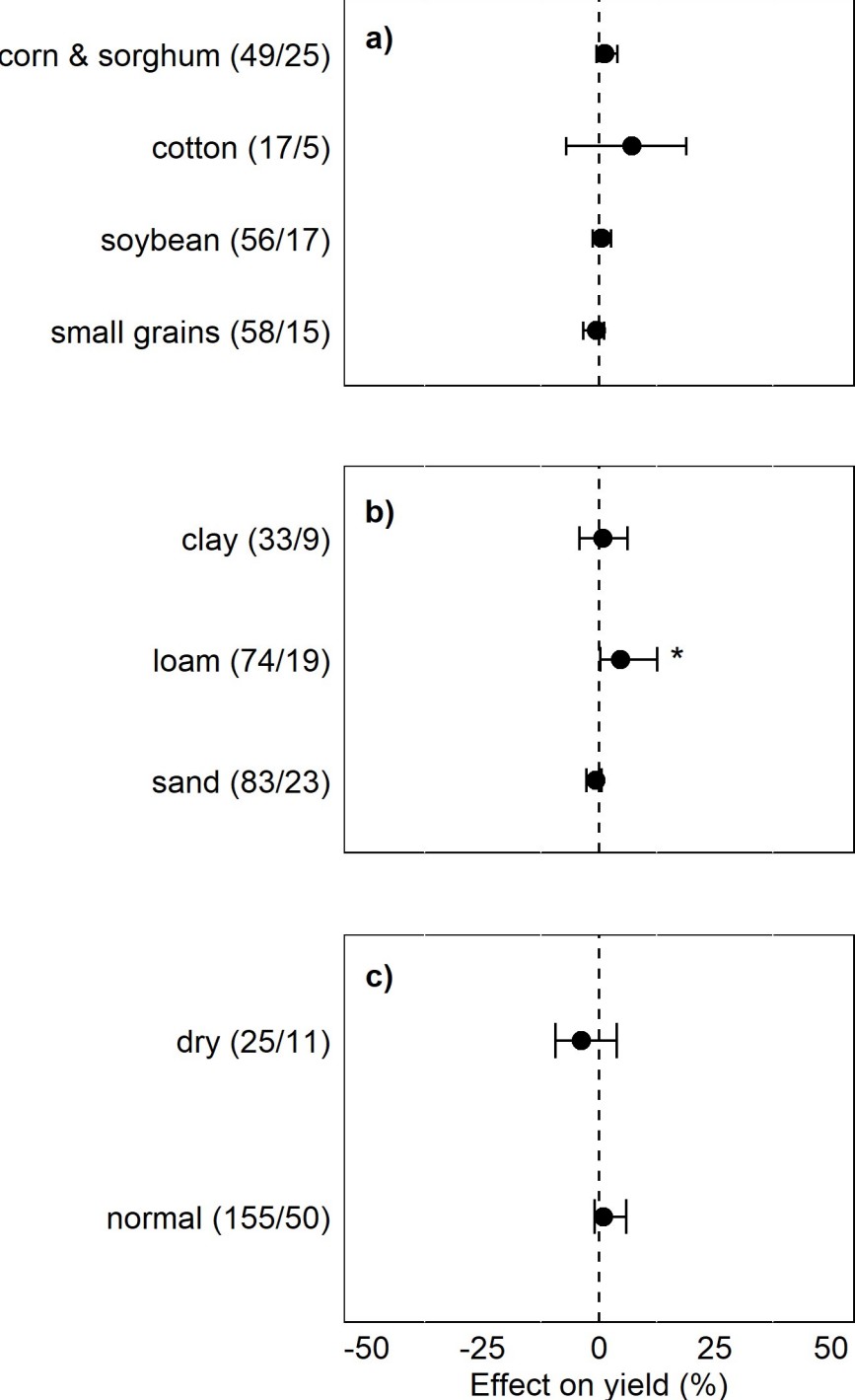

**Fig 5.** Effect of ICLS on crop yield relative to unintegrated systems within subgroups for (A) crop type, (B) soil texture, and (C) during dry or normal precipitation years. Excludes observations from dual-purpose cropping systems. Number of observations/number of studies for each category appears in parentheses. Categories with less than 15 observations were omitted from the subgroup analysis, as were observations from dual-purpose cropping systems. Points represent grazed system yield effect, while the dotted vertical line represents the paired ungrazed system yield. Error bars represent 95% bias-corrected-accelerated bootstrap confidence intervals. Asterisks (*) represent a significant yield response in grazed systems relative to ungrazed systems at the 95% confidence level.

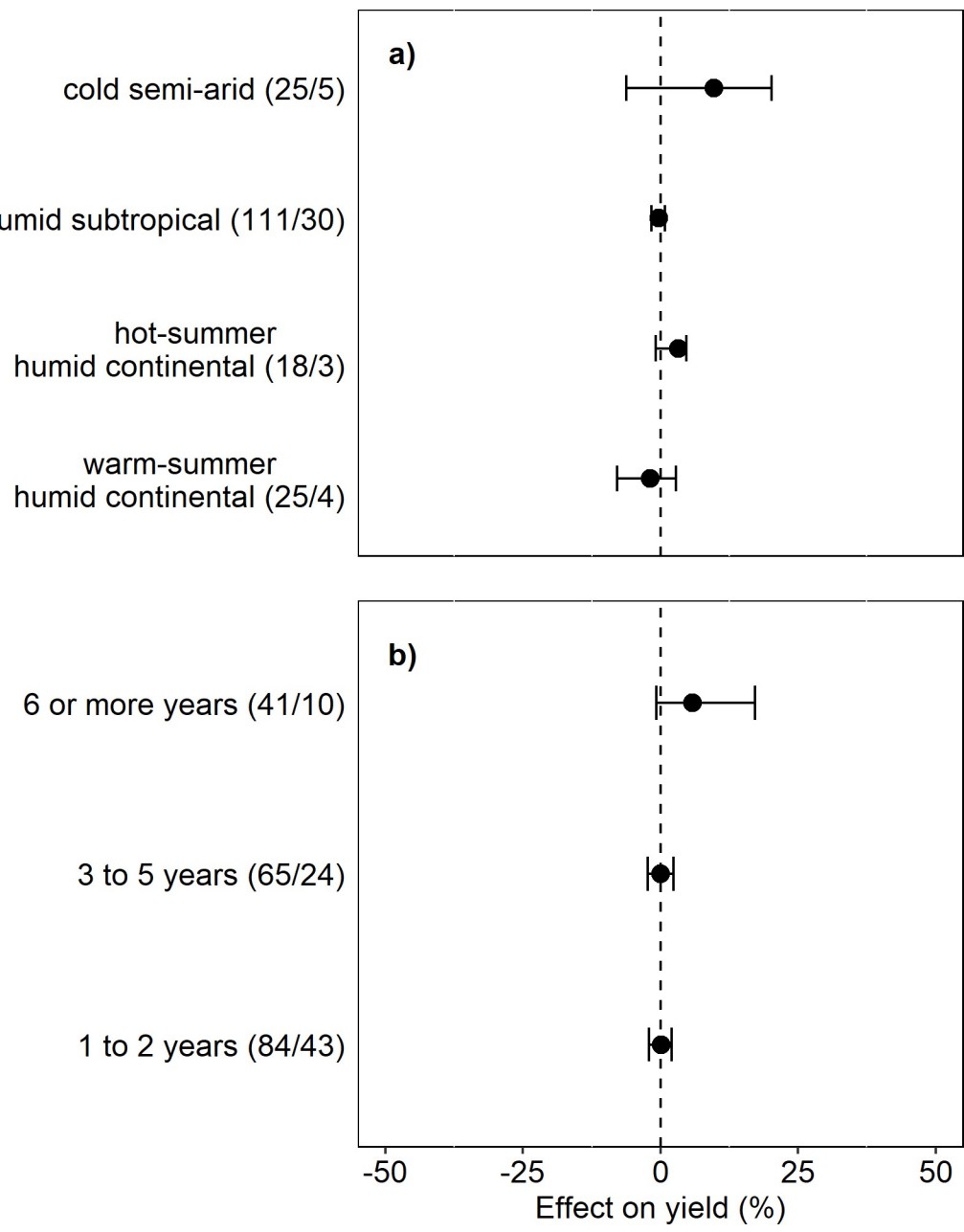

**Fig 6.** Effect of ICLS on crop yield relative to unintegrated systems within subgroups for (A) climate and (B) length of study. Excludes observations from dual-purpose cropping systems. Climate categories are derived from the Köppen classification system. Number of observations/number of studies for each category appears in parentheses. Categories with less than 15 observations were omitted from the subgroup analysis. Points represent grazed system yield effect, while the dotted vertical line represents the paired ungrazed system yield. Error bars represent 95% bias-corrected-accelerated bootstrap confidence intervals. Asterisks (*) represent a significant yield response in grazed systems relative to ungrazed systems at the 95% confidence level.

## Discussion

We applied meta-analytic methods to 66 studies to understand the impact of commercial crop-livestock integration on crop production. In addition, we examined the effect of environmental context and other moderating variables on productive outcomes in integrated relative

to unintegrated systems. We found that when grazing is included in cropping system design, average crop yields are the same as yields in ungrazed systems across a wide variety of environmental and management contexts. It should be emphasized that, in terms of outcomes for ecological intensification of agriculture and improvements in land-use efficiency, a non-effect of livestock integration on crop yield is as beneficial as a positive effect. Multi-enterprise systems contribute not only to increased whole-system economic and agronomic output, but to improved ecosystem function via biodiversity [106,107] and land-sparing benefits [108]. In other words, successful ICLS–especially ICLS that do not increase input use relative to non-integrated systems–can generate more product per unit of land area or input, thereby reducing the need for agricultural expansion into intact native ecosystems.

The overall neutral effect of ICLS systems on crop yields is consistent with reports in the literature, many of which report no significant effect of grazing on subsequent crop yield but note other benefits such as whole-system economic productivity, soil health, and water conservation [29,31,35]. Some studies report significantly negative effects of grazing on subsequent crop yield, although typically only when conducted under sub-optimal conditions such as extremely wet or recently thawed soils (e.g., [80]). Also, because of the complexity of designing and conducting experiments with both crop and livestock components, some negative results have been attributed to improper methodological conditions such as lack of pasture fertilization and/or inappropriate stocking rates [10]. These observations underscore the importance of management strategies that are tailored to individual environmental and system contexts and that carefully observe best management practices.

We observed no production penalty for crops grown in rotation with a grazing treatment (cover crop grazing, forage/sod rotation, or stubble grazing). These findings come at a time when yield gains for most major grain crops are slowing, climate change is increasingly impacting yield potentials, and crop breeders are struggling to find new avenues for large yield gains under both limiting and optimal conditions [109]. Furthermore, rising concerns about land conversion to agriculture and pasture in sensitive or important ecosystems such as the Brazilian Amazon [5,6,110] are stoking interest in identifying viable strategies for ecological intensification of existing agricultural land area. Therefore, system designs and management techniques that contribute to closing yield gaps while generating multiple other ecological and economic benefits are critical. Our results suggest that given reasonable market and policy environments [14], and assuming use of best grazing management practices, grazing can be coupled with crop production to generate increases in productivity per unit land area without great risk of compromising crop yields [80].

The yield penalty observed for dual-purpose crops compared to ungrazed, single-purpose crops in this analysis points to the unique challenges involved in this ICLS. Managers of dual-purpose crops must strike a delicate balance among crop variety selection, grazing timing, grazing intensity, and acceptable levels of soil compaction under largely unpredictable weather conditions. Crops grazed in the vegetative stage may outyield ungrazed crops under specific circumstances, such as when a late frost is avoided because of the delay in flowering caused by grazing, or when a dry season rewards systems that conserve soil water through early-season control of vegetative biomass [58]. However, on most other occasions, a small yield loss due to grazing the vegetative stage of a crop is to be expected. Once again, the whole-system benefits of dual-purpose grazing should be emphasized over yield-centric outcomes. The analysis here compares systems based on yield alone, which may not capture benefits or tradeoffs accruing from the wider system. For example, producers may be more than compensated for yield losses from grazed dual-purpose crops by the farm revenue gained from the additional livestock production value [86]. Dual-purpose managers further state the advantages of grazable crops that allow them to rest pastures, and thus rehabilitate and extend pasture lifespan [111]. Similarly, a

grazable crop can be sacrificed if conditions appear more favorable for livestock production than crop production, giving producers greater adaptability in the face of unexpected climate events [27].

Crop production outcomes in ICLS were relatively unaffected by crop or livestock categories except when observations from dual-purpose systems were included, in which case the effect was negative for canola and small grains (wheat, in particular). The effect of ICLS on canola yield could not be tested with the exclusion of dual-purpose crop observations due to the small sample size of canola observations in other kinds of ICLS.

A striking result of this meta-analysis is that relationships between crop production outcomes in ICLS relative to unintegrated systems were largely unaffected by climate- and weather-related factors. ICLS performed similarly to unintegrated systems in all climate categories except for humid subtropical, a category which again represented a disproportionate number of dual-purpose cropping studies. Furthermore, we observed no effect of livestock integration on crop yield in either dry or normal precipitation years. Crop-livestock integration did not appear to negatively affect crop ability to withstand dry weather-related stress, despite reports of animal grazing lowering soil water content in some contexts [112].

The non-effect of ICLS on yields in clayey and sandy soils and the positive effect in loams (when dual-purpose crops were excluded) is consistent with model simulations showing that soil compaction due to animal traffic is usually too mild to affect subsequent crop growth and is often ameliorated by root action or tillage at depth [43]. Many studies have shown little to no effect of animal grazing on soil physical properties over the long term, despite concerns voiced by growers about compaction [113,114]. It is also important to note that only the optimal grazing management treatments from each study were included in this analysis. ICLS conducted at higher grazing intensities could easily result in negative outcomes from livestock integration due to degradation in soil quality [115].

The positive effect of ICLS in loamy soils suggests that intermediate soil textures may represent a "sweet spot" for ICLS implementation. The reasons for the positive effect of ICLS in intermediate soil textures potentially include the physical and hydrological properties of loams, which are more resistant to compaction than clayey soils and more robust against organic carbon loss, erosion, and drought impacts than sandy soils [6]. They are also considered among the most conducive to no-till management, where the benefits of adding grazing animals to the system are more likely to be seen in the form of acidity amelioration and other soil chemical improvements [24] along with increased soil carbon accumulation rates [115].

The slightly negative effect of ICLS in sandy soils cannot be attributed to the presence of observations from dual-purpose crops alone, as these studies were relatively equally represented in both sandy and loamy soils. Although this effect was not significant when dual-purpose crop observations were excluded, the negative tendency suggests greater sensitivity of sandy soils to sub-optimal conditions. For example, Andrés et al. [116] showed that C respiration and N mineralization in fine-textured soils were unaffected by rainfall pattern or grazing management, but that both rates were significantly lower in coarse-textured soils under an altered rainfall pattern. This effect was exacerbated by grazing activity. Conversely, the chemical properties of fine-textured soils such as pH, cation exchange capacity, and extractable Ca + and Mg+ can be more responsive than coarse-textured soils to the effects of moderate grazing, an attribute that may convey more robustness to environmental stressors in fine-textured soils [24].

When observations from dual-purpose systems were excluded, there was no effect of study duration on ICLS performance relative to unintegrated systems. When dual-purpose system observations were included, yield penalties associated with animal grazing were seen in studies of less than 2 years in duration. However, this result is likely related to the shorter duration on

average of studies involving dual-purpose cropping systems. The effects of crop-livestock integration are typically expected to occur after a delay, as most direct impacts of animal grazing are acting on soil physical and chemical properties that are relatively slow to change [114]. Furthermore, gains in manager experience may be just as important as gains/losses in soil quality when explaining relative yields in transitioning systems, as demonstrated in studies of conventional-to-organic transitions [117]. In this sense, dual-purpose cropping systems may benefit from increased manager experience over longer time periods. As for other ICLS, our results suggest that there is no "break-in" period for systems in transition to ICLS and that well-managed, newly implemented ICLS perform as well as long-term ICLS relative to unintegrated systems.

Limitations common to ICLS studies should be taken into account when interpreting meta-analysis results. For example, many such studies involve trials with low replication– 3 or 4 replicates per treatment–due to the large land areas required for grazing and farming system trials. Although within-study standard errors of the studies included in this meta-analysis were much lower than among-study heterogeneity, within-study error due to low replication cannot be disregarded. Furthermore, meaningful multilevel meta-analysis of the effects of fertilizer inputs and application rates, as well as the effects of grazing management (stocking densities, forage allowances, etc.), were limited both by deficiencies in the published data and by the complexity of interacting environmental factors. Animal stocking densities, for example, are often not reported in ICLS studies, and when they are, units of measure are inconsistent and difficult to transform into a common measure. Furthermore, optimal stocking rates are dependent on forage type, season, and animal weight group, among others, meaning that objectively comparing stocking densities among study sites can be challenging.

The results of meta-analysis should therefore be interpreted with care. ICLS outcomes are undoubtedly contingent on the use of best grazing management practices such as appropriate stocking rates and timing of management operations [115]. Furthermore, important management treatments such as nitrogen application rates, tillage types, etc., were held constant in this analysis, but could offer opportunities for future studies to investigate their impact on the outcome of crop production in ICLS. It is important to note that the approach adopted here allows only for within-system comparisons, e.g., comparison between two soybean production systems that are identical except for their integration or not of grazed cover crops. Cross-system comparisons cannot be made without significantly broadening the scope to include information on socioeconomic and environmental similarities/dissimilarities. A broader range of ICLS types (including smallholder and mixed farming systems), scales (including cross-farm and territorial level ICLS), and geographies would also be warranted for future study as ICLS research continues to expand. Socio-economic and political factors at play in different regions will also impact whether ICLS results in positive or negative outcomes beyond crop yields alone due to the counter-balancing effects of risk mitigation through diversified revenue streams, increased managerial complexity of ICLS, and incentives or disincentives to ICLS adoption in a given policy environment. In any case, this study represents an important first step in collating diverse studies and assessing the key moderating variables contributing to ICLS outcomes in three major global production regions.

## Conclusion

Our results clearly show the potential of ICLS as an ecological intensification strategy. Meta-analysis of ICLS across 5 climates, 3 broad soil textures, 12 crops, and 4 livestock species showed that livestock integration has no impact on crop yields in large scale industrialized systems. Exceptions were crops and climates involved in dual-purpose cropping systems (canola

and wheat; humid subtropical climate). However, differences in response among categories within crop, livestock, climate, and soil texture subgroups were minimal and in most other cases crop yields were unaffected by livestock integration. This study represents the first time that crop production outcomes in ICLS have been examined quantitatively across studies conducted in different regions and system types, and it is the first example to synthesize ICLS outcomes across significant regional variation in management scenarios. These complex systems warrant continued research in a variety of contexts to increase our understanding beyond yield outcomes to ecological, agronomic, and economic outcomes as well, and their diverse underlying drivers and mechanisms.

## Supporting information

**S1 Checklist. PRISMA checklist 2009.**
(DOC)

**S1 Fig. Funnel plot illustrating the trim and fill method used to detect potential plot asymmetry indicating publication bias [96].** Black points represent the mean effect size of each study included in the meta-analysis. White points are hypothetical missing studies infilled by an algorithm to achieve symmetry in the funnel plot.
(DOCX)

**S2 Fig.** Effect of ICLS on crop yield relative to unintegrated systems within subgroups for a) crop type, b) soil texture, and c) during dry or normal precipitation years. Includes observations from dual-purpose cropping systems. Number of observations/number of studies for each category appears in parentheses. Categories with less than 15 observations were omitted from the subgroup analysis, as were observations from dual-purpose cropping systems. Points represent grazed system yield effect, while the dotted vertical line represents ungrazed system yields. Error bars represent 95% bias-corrected-accelerated bootstrap confidence intervals. Asterisks (*) represent a significant yield response in grazed systems relative to ungrazed systems at the 95% confidence level.
(DOCX)

**S3 Fig.** Effect of ICLS on crop yield relative to unintegrated systems within subgroups for a) climate and b) length of study. Includes observations from dual-purpose cropping systems. Climate categories are derived from the Köppen classification system. Number of observations/number of studies for each category appears in parentheses. Categories with less than 15 observations were omitted from the subgroup analysis. Points represent grazed system yield effect, while the dotted vertical line represents ungrazed system yields. Error bars represent 95% bias-corrected-accelerated bootstrap confidence intervals. Asterisks (*) represent a significant yield response in grazed systems relative to ungrazed systems at the 95% confidence level.
(DOCX)

**S4 Fig.** Effect of ICLS on crop yield relative to unintegrated systems within subgroups for a) livestock type with dual-purpose crop studies excluded and b) livestock type with dual-purpose crop studies included. Number of observations/number of studies for each category appears in parentheses. Categories with less than 15 observations were omitted from the subgroup analysis. Points represent grazed system yield effect, while the dotted vertical line represents ungrazed system yields. Error bars represent 95% bias-corrected-accelerated bootstrap confidence intervals. Asterisks (*) represent a significant yield response in grazed systems relative to ungrazed systems at the 95% confidence level.
(DOCX)

## Acknowledgments

The authors would like to acknowledge significant contributions from PCF Carvalho, A Moraes, and T Kunrath for assistance in data collection and contacting study authors.

## Author Contributions

**Conceptualization:** Caitlin A. Peterson, Leonardo Deiss, Amélie C. M. Gaudin.

**Data curation:** Caitlin A. Peterson, Amélie C. M. Gaudin.

**Formal analysis:** Caitlin A. Peterson, Leonardo Deiss.

**Funding acquisition:** Caitlin A. Peterson, Amélie C. M. Gaudin.

**Investigation:** Caitlin A. Peterson.

**Writing – original draft:** Caitlin A. Peterson.

**Writing – review & editing:** Caitlin A. Peterson, Leonardo Deiss, Amélie C. M. Gaudin.

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
