## [Decision Letter · Decision Letter 0]

20 Feb 2020

PONE-D-19-36052

Integrated Crop-Livestock Systems achieve comparable crop yields to specialized systems: a meta-analysis

PLOS ONE

Dear Dr. Peterson,

Thank you for submitting your manuscript to PLOS ONE. After careful consideration, we feel that it has merit but does not fully meet PLOS ONE’s publication criteria as it currently stands. Therefore, we invite you to submit a revised version of the manuscript that addresses the points raised during the review process.

The manuscript is wwell written and to the point and I feel it takes few corrections to be acceptable in Plos1. Please answer the comments given by the reviewers, which were consistent each other.

We would appreciate receiving your revised manuscript by Apr 05 2020 11:59PM. To enhance the reproducibility of your results, we recommend that if applicable you deposit your laboratory protocols in protocols.io, where a protocol can be assigned its own identifier (DOI) such that it can be cited independently in the future. For instructions see: http://journals.plos.org/plosone/s/submission-guidelines#loc-laboratory-protocols

We look forward to receiving your revised manuscript.

Kind regards,

Sergio Saia, Ph.D.

Academic Editor

PLOS ONE

Additional Editor Comments (if provided):

Just 3 not important suggestion:

L82 change "acidity" to "soil pH"

L104: please give some hints on the "adverse".

L106: change "moderating variables governing production" to "variables related to the"

Journal Requirements:

Reviewers' comments:

Reviewer's Responses to Questions

**Comments to the Author**

1. Is the manuscript technically sound, and do the data support the conclusions?

Reviewer #1: Yes

Reviewer #2: Yes

Reviewer #3: Yes

2. Has the statistical analysis been performed appropriately and rigorously? 

Reviewer #1: Yes

Reviewer #2: Yes

Reviewer #3: I Don't Know

3. Have the authors made all data underlying the findings in their manuscript fully available?

Reviewer #1: Yes

Reviewer #2: Yes

Reviewer #3: No

4. Is the manuscript presented in an intelligible fashion and written in standard English?

Reviewer #1: Yes

Reviewer #2: Yes

Reviewer #3: Yes

5. Review Comments to the Author

Reviewer #1: PLOSONE

REVIEWER COMMENTS

MS No. PONE-D-19-36052_reviewer

Title: Integrated Crop-Livestock Systems achieve comparable crop yields to specialized systems: a meta-analysis

COMMENTS

The manuscript argues a study addressed to three main objectives:

- understand the effect of crop-livestock integration on crop productivity under normal and adverse weather conditions;

- determine likely production outcomes across environments and integrated crop-livestock systems types;

- identify the most important moderating variables governing production outcomes for each type of integrated crop-livestock systems.

The topic and the objectives of the study are very interesting from scientific point of view, therefore stimulated me to a careful reading.

The manuscript is well structured and organized, according to the journal directions (Introduction; Methods; Results; Discussion; Conclusions). Each part of the manuscript is well written and understandable.

Nevertheless, I found some little faults. So, I reported below some observations and considerations.

I understand that the Authors included in the meta-analysis the studies that met specific criteria, but the fact that the study was only concentrate on North America, South America, and Australia determines a little limitation.

In the methods, the Authors specified that crop species were grouped according to broad agronomic similarities and animals were grouped into small ruminants and cattle. So, what about the grouping for climate type, such as Mediterranean and continental? This aspect needs to be included to sustain the considerations reported in the discussion and conclusion.

In figure 3, 5 and 6: "the dotted vertical lines represent ungrazed system yields"; I see just a vertical line at zero, where are the others?

Figure 3: the error bars in dual-purpose crop and in forage rotation that are much larger than others is a little bit strange, please check them.

The analysis that considered the dry or normal precipitation years (such as that reported in fig 5) could be absolutely correct but must be referred to the climate type. This because 500 mm can be normal precipitation in the Mediterranean climate and at the same time 1000 mm can be normal precipitation in the continental climate. I would like to suggest the distinction by climate type.

So, I would suggest that a little improvement with minor modifications of the manuscript is needed for publish it on PLOSONE.

RECOMMENDATION:

I suggest MINOR revisions to the manuscript before the accepting it.

Reviewer #2: This paper adds some interesting insights, supported by quantitative data from a systemic meta-analysis, to the evidence base on integrated crop livestock systems operated at medium and large scale. I read it with interest. Smallholder crop livestock systems were not included which is not a shortcoming but should be highlighted a little more strongly when describing the literature surveyed; and even in the title perhaps.

The study itself, quite legitimately given the nature of the studies used in the meta-analysis, uses only crop yield as an outcome variable. This leads to a number of difficulties in interpretation. Why on earth would any rational farm manager opt for dual purpose crops on the basis of this analysis? This is easily resolved by clearly acknowledging that crop yield is not the best outcome variable for evaluating such systems “holistically”. There are economically valuable outputs from the livestock components which mean that crop yield and total system output do not correlate well (e.g. for dual purpose crops). At present I can only see this being hinted at in line 406 and the impression left is that it is only eco-system services that justify the inclusion of livestock and not the tomahawk steaks. Not having the data to evaluate this trade off does not kill this paper, in my view, as the implications for crop yield and soil factors and the discussions around these are important and well-articulated. This issue should, however be alluded to in relevant sections. Related to this, using the term “broad production outcomes” in the abstract is a little difficult to justify!

Reviewer #3: This is a carefully conducted and well written meta-analysis on yields of integrated (crop-livestock) cropping systems as compared to arable cropping alone.

I trust that the statistical analysis is well done – personally I cannot judge it and I hope that another reviewer will.

I have just a couple of comments.

1) “Crop production (…) remains uncertain and a barrier to adoption” (line 16). Why is this? Are really yields the major barrier to adoption? Based on my (European) experience, I would not have this hypothesis. In my experience, the trend towards specialized cropping – and the inexistence of a trend towards ICLS – is due to organizational aspects and labour costs, mostly. Not crop yield, but – if any – the yield performance of livestock systems.

2) “Studies (…) did not include application of manure as slurry or compost” (line 205). The application of organic manure is a key feature of integrated farming systems – at least in my experience. Did they really not include organic manure application – or just not report about it?

3) The four types of ICLS are not fully self explaining. I would appreciate a short description (a sentence or two) describing each of the four strategies, and also the control systems that they were compared to. This could even be a table, instead of text. Please also briefly elaborate on the livestock type. I only realized at line 308 that small ruminants were also included. Were these many studies? In particular regions?

4) I would probably make sense, in the discussion, to more explicitly point out that the study does not compare the systems but only looks at the specific aspect of crop yield. To compare the systems, also livestock performance would have to be evaluated and the two “yields” would need to be evaluated together. In addition, other aspects would also need to be evaluated (environmental, economic, social). This is beyond the scope of the article, of course, and it is implicitly also mentioned. I would just summarize it in two or three sentences – as this is the kind of analysis that would actually be needed.

And: Figure 5 and 6 x-Axis: do they really have to scale to 100%? Rather scale to 50% or 25%?

6. PLOS authors have the option to publish the peer review history of their article (what does this mean?). If published, this will include your full peer review and any attached files.

Reviewer #1: No

Reviewer #2: Yes: Peter Thorne

Reviewer #3: No

---

## [Author Response · Author response to Decision Letter 0]

16 Mar 2020

Reviewer #1 COMMENTS

The manuscript argues a study addressed to three main objectives:

- understand the effect of crop-livestock integration on crop productivity under normal and adverse weather conditions;

- determine likely production outcomes across environments and integrated crop-livestock systems types;

- identify the most important moderating variables governing production outcomes for each type of integrated crop-livestock systems.

The topic and the objectives of the study are very interesting from scientific point of view, therefore stimulated me to a careful reading.

The manuscript is well structured and organized, according to the journal directions (Introduction; Methods; Results; Discussion; Conclusions). Each part of the manuscript is well written and understandable.

Nevertheless, I found some little faults. So, I reported below some observations and considerations.

I understand that the Authors included in the meta-analysis the studies that met specific criteria, but the fact that the study was only concentrate on North America, South America, and Australia determines a little limitation. 

RESPONSE: We agree that the geographic representation of studies included in the analysis is very limited. However, we believe it is an accurate reflection of the available literature meeting our requirements. Altering our inclusion criteria so as, for example, to include smallholder operations practicing whole-farm crop-livestock integration would fundamentally change the nature and scope of our analysis. Furthermore, systems would be so different from one another in their objectives (e.g., subsistence vs. commercialization) and limitations (e.g. socioeconomic limitations vs. agronomic limitations) as to prevent drawing realistic comparisons. This is especially true given that we used crop yield outcomes as our metric of system performance. Comparing yield outcomes in fully industrialized operations to yields in resource-limited smallholder settings would result in systematic biases. We further justify our approach in L155-158 regarding inclusion criteria. A second, separate study examining smallholder integrated systems would be more informative than a single study incorporating disparate geographies and system typologies. 

In the methods, the Authors specified that crop species were grouped according to broad agronomic similarities and animals were grouped into small ruminants and cattle. So, what about the grouping for climate type, such as Mediterranean and continental? This aspect needs to be included to sustain the considerations reported in the discussion and conclusion. 

RESPONSE: Earlier in this paragraph (L209-220) we also state that information on climate and soil classes for the geographic location of each study were extracted from the relevant databases. Climate class was assigned to each study using the Koppen classification system, e.g., tropical savannah, hot semi-arid, cold semi-arid, humid subtropical, among others. These classifications are based on annual temp and precipitation patterns. 

In this revised version of the manuscript we have omitted the mention of continuous temperature and precipitation variables being extracted from the WorldClim database for each study, as these variables were not ultimately utilized in the analysis.

In figure 3, 5 and 6: "the dotted vertical lines represent ungrazed system yields"; I see just a vertical line at zero, where are the others? 

RESPONSE: Our figure caption was misleading here; they should refer only to the single dotted line visible at 0 on the x-axis. Wording has been changed to correct this.

Figure 3: the error bars in dual-purpose crop and in forage rotation that are much larger than others is a little bit strange, please check them. 

RESPONSE: We have re-run the bootstrap CI analysis for ICLS categories and the upper and lower confidence intervals are correct as represented in Figure 3. We suggest that the larger CI for forage rotation may be due to the smaller sample size for that ICLS type (n=19, compared to n=56, n=70, and n=101 for dual-purpose crop, stubble grazing, and cover crop grazing, respectively), while the CI for dual-purpose crop may be due to the high degree of both among- and within-study variability in yield outcomes for grazed crops, from total crop failure to slight improvements in yield relative to single-purpose crops.

The analysis that considered the dry or normal precipitation years (such as that reported in fig 5) could be absolutely correct but must be referred to the climate type. This because 500 mm can be normal precipitation in the Mediterranean climate and at the same time 1000 mm can be normal precipitation in the continental climate. I would like to suggest the distinction by climate type.

The studies included in the dry/normal precipitation years analysis inherently considered climate type because we used the author’s assessment of weather anomalies rather than absolute thresholds for mm rainfall across all study locations. For example, what authors designated as an unusually dry year for a study in NSW Australia would differ in absolute rainfall from what authors designated a dry year in Texas, USA. This approach is noted in L222-224. 

Unfortunately, we are unable to test the interaction between climate group and climate anomaly years for its potential effect on crop yield because most of the subgroups would not have a large enough sample size to perform the bootstrap analysis. Studies in the hot-summer humid continental, temperate oceanic, tropical savannah, and hot semi-arid climate groups did not include any observations of dry anomaly years, while studies in the warm-summer humid continental and cold semi-arid climate groups had only 1 and 7 observations during dry anomaly years, respectively. There was only one interaction (humid subtropical x dry anomaly) with enough observations to perform the bootstrapping procedure, which would not make for an interesting comparison.

Reviewer #2 COMMENTS: 

This paper adds some interesting insights, supported by quantitative data from a systemic meta-analysis, to the evidence base on integrated crop livestock systems operated at medium and large scale. I read it with interest. Smallholder crop livestock systems were not included which is not a shortcoming but should be highlighted a little more strongly when describing the literature surveyed; and even in the title perhaps. 

RESPONSE: We agree that is important to emphasize the distinction made between commercially oriented and smallholder ICLS modalities in this study. We have expanded our justification for excluding smallholder ICLS modalities in L77-86. We have also changed the title to reflect this narrower definition of ICLS (i.e. commercial only): “Commercial ICLS achieve comparable yields to specialized production systems: a meta-analysis.”

The study itself, quite legitimately given the nature of the studies used in the meta-analysis, uses only crop yield as an outcome variable. This leads to a number of difficulties in interpretation. Why on earth would any rational farm manager opt for dual purpose crops on the basis of this analysis? This is easily resolved by clearly acknowledging that crop yield is not the best outcome variable for evaluating such systems “holistically”. There are economically valuable outputs from the livestock components which mean that crop yield and total system output do not correlate well (e.g. for dual purpose crops). At present I can only see this being hinted at in line 406 and the impression left is that it is only eco-system services that justify the inclusion of livestock and not the tomahawk steaks. Not having the data to evaluate this trade off does not kill this paper, in my view, as the implications for crop yield and soil factors and the discussions around these are important and well-articulated. This issue should, however be alluded to in relevant sections. 

RESPONSE: We completely agree with this assessment! Yield outcomes should not be the only basis on which to judge the overall productivity of diversified systems. In fact, even small losses in yield due to livestock activity may by more than compensated by the additional revenues brought in by the livestock component. This would be a justification for farmers to implement dual-purpose, for example, despite the likely yield losses, in addition to the management flexibility they would gain to sacrifice the crop in favor of livestock fodder if the year looks to be poor for grains. We have added additional text in L435-435 and L534 to this effect.

Related to this, using the term “broad production outcomes” in the abstract is a little difficult to justify! 

RESPONSE: We have changed “broad production outcomes” to “crop production outcomes” (now L36 in the abstract) to more accurately reflect the scope of the analysis.

Reviewer #3 COMMENTS: 

This is a carefully conducted and well written meta-analysis on yields of integrated (crop-livestock) cropping systems as compared to arable cropping alone.

I trust that the statistical analysis is well done – personally I cannot judge it and I hope that another reviewer will.

I have just a couple of comments.

1) “Crop production (…) remains uncertain and a barrier to adoption” (line 16). Why is this? Are really yields the major barrier to adoption? Based on my (European) experience, I would not have this hypothesis. In my experience, the trend towards specialized cropping – and the inexistence of a trend towards ICLS – is due to organizational aspects and labour costs, mostly. Not crop yield, but – if any – the yield performance of livestock systems. 

RESPONSE: This is very true and a valid point. However, we do not state that yield losses are the major barrier to production, merely one barrier among several. We have found that when producers express hesitancy about implementing an integrated system they generally cite direct biophysical factors of concern, such as soil compaction leading to reduced crop yields, nutrient depletion from forage production that is then exported as animal products, or leaving soil exposed and subject to erosion by consuming crop residues. However, most of these problems are due to grazing mismanagement, something which we wish to highlight here by showing the largely neutral effect of animal grazing on crop production.

As you say, organizational, economic, and labor issues are also clearly a barrier, in both the European and North American contexts, and we do not wish to belie this point. But these barriers have been explored extensively by other authors (see for example Garrett et al. 2017, Moraine, Duru and Therond 2016 etc.). Our objective with this study was to provide one more piece to the puzzle, rather than an exhaustive view of barriers to adoption.

L109-110 makes reference to this previous literature. Also, the line in the abstract referred to here has been modified to reflect that the barrier to adoption we address here is only one among many (L27-28 in the revised manuscript).

2) “Studies (…) did not include application of manure as slurry or compost” (line 205). The application of organic manure is a key feature of integrated farming systems – at least in my experience. Did they really not include organic manure application – or just not report about it?

RESPONSE: The reviewer is correct that manure applications are a common feature of territory-scale ICLS, such as in the European context. Territory-scale ICLS refers to integration that occurs among different farms within a territory that cooperate to supply needed inputs, e.g. grazing land for livestock or manure for crop nutrient management. 

As stated in item 3 of the inclusion criteria (L166), the present study includes only field-scale ICLS, which refers to integration that is co-located within the same land area, e.g. cattle being grazed and crops being grown on the same parcel. These ICLS scales are distinguished in order to isolate the potential effect of grazing animals on arable crop land, rather than the broader socio-economic considerations that are important in territory-scale and even farm-scale ICLS. The use of this criterium also implies that cut-and-carry type systems or systems implementing manure applications rather than direct livestock grazing on cropland were excluded.

This distinction may also explain why many studies from European geographies were excluded from the meta-analysis, as the territory-scale definition is more commonly used there than the field-scale definition of ICLS.

Text was added to L182-187 in the methods to this effect. Co-located ICLS categories examined in the meta-analysis are described in L177-181 and in Table 2.

3) The four types of ICLS are not fully self explaining. I would appreciate a short description (a sentence or two) describing each of the four strategies, and also the control systems that they were compared to. This could even be a table, instead of text. Please also briefly elaborate on the livestock type. I only realized at line 308 that small ruminants were also included. Were these many studies? In particular regions? 

RESPONSE: Categories of co-located ICLS are described in L177-181. As suggested, we have also added a table to clarify distinctions among ICLS types and to better explain the control systems they were compared to (Table 2).

The livestock type examined in each study is noted in Table 1 in the column headed “Animal.” We now also make parenthetical reference to the livestock types examined in L176.

4) I would probably make sense, in the discussion, to more explicitly point out that the study does not compare the systems but only looks at the specific aspect of crop yield. To compare the systems, also livestock performance would have to be evaluated and the two “yields” would need to be evaluated together. In addition, other aspects would also need to be evaluated (environmental, economic, social). This is beyond the scope of the article, of course, and it is implicitly also mentioned. I would just summarize it in two or three sentences – as this is the kind of analysis that would actually be needed. 

RESPONSE: Yes, only within-system comparisons are made, i.e., how does a soybean production system compare to another soybean production system with the same characteristics but with livestock grazing integrated?

This is an excellent point, and we have included additional text in the future research section L515-519 to elaborate on the limitations of the scope and the additional requirements that would need to be met to enable cross-system comparisons. Other approaches should be used to evaluate the systems including social, economic, and environmental. But it is worth noting that in the context of sustainable intensification, having similar crop yields plus animal production can reduce pressure to convert natural ecosystems to sustain agricultural production for a growing population.

And: Figure 5 and 6 x-Axis: do they really have to scale to 100%? Rather scale to 50% or 25%?

RESPONSE: All forest plot figure x-axes (including figures in supporting information) are now scaled to 50%.

---

## [Editor Report · Decision Letter 1]

2 Apr 2020

Commercial Integrated Crop-Livestock Systems achieve comparable yields to specialized production systems: a meta-analysis

PONE-D-19-36052R1

Dear Dr. Peterson,

We are pleased to inform you that your manuscript has been judged scientifically suitable for publication and will be formally accepted for publication once it complies with all outstanding technical requirements.

With kind regards,

Sergio Saia, Ph.D.

Academic Editor

PLOS ONE

---

## [Editor Report · Acceptance letter]

27 Apr 2020

PONE-D-19-36052R1 

Commercial Integrated Crop-Livestock Systems achieve comparable crop yields to specialized production systems: a meta-analysis 

Dear Dr. Peterson:

I am pleased to inform you that your manuscript has been deemed suitable for publication in PLOS ONE. Congratulations! Your manuscript is now with our production department. 

With kind regards,

on behalf of

Dr. Sergio Saia 

Academic Editor

PLOS ONE